# Antibodies against SARS-CoV-2 Alpha, Beta, and Gamma Variants in Pregnant Women and Their Neonates under Antenatal Vaccination with Moderna (mRNA-1273) Vaccine

**DOI:** 10.3390/vaccines10091415

**Published:** 2022-08-28

**Authors:** Wei-Chun Chen, Yen-Pin Lin, Chao-Min Cheng, Ching-Fen Shen, Alex Ching, Ting-Chang Chang, Ching-Ju Shen

**Affiliations:** 1Institute of Biomedical Engineering, National Tsing Hua University, Hsinchu 300, Taiwan; 2Division of Gynecologic Oncology, Department of Obstetrics and Gynecology, Chang Gung Memorial Hospital at Linkou, College of Medicine, Chang Gung University, Taoyuan 333, Taiwan; 3Department of Obstetrics and Gynecology, New Taipei City Municipal Tucheng Hospital, New Taipei City 236, Taiwan; 4Department of Pediatrics, National Cheng Kung University Hospital, College of Medicine, National Cheng Kung University, Tainan 701, Taiwan; 5Department of Materials Science and Engineering, Carnegie Mellon University, Pittsburgh, PA 15123, USA; 6Department of Obstetrics and Gynecology, Kaohsiung Medical University Hospital, Kaohsiung Medical University, Kaohsiung 807, Taiwan

**Keywords:** COVID-19 vaccine, Alpha-type SARS-CoV-2, Beta-type SARS-CoV-2, Gamma-type SARS-CoV-2, SRBD IgG, influenza vaccine, pertussis vaccine, Tdap vaccine

## Abstract

The aim of the study was to examine the impact of COVID-19 vaccination on the anti-SARS-CoV-2 spike receptor binding domain IgG antibody (SRBD IgG) binding ratio (SBR) from Alpha, Beta, and Gamma variants of SARS-CoV-2 in pregnant women and neonates. The impact of antenatal influenza (flu) and pertussis (Tdap) vaccines was also studied. We enrolled pregnant women vaccinated with the Moderna (mRNA-1273) vaccine during pregnancy and collected maternal plasma (MP) and neonatal cord blood (CB) during delivery to determine the SBR via enzyme-linked immunosorbent assays (ELISA). A total of 78 samples were collected from 39 pregnant women. The SBR was higher for Alpha variants compared to Beta/Gamma variants (MP: 63.95% vs. 47.91% vs. 43.48%, *p* = 0.0001; CB: 72.14% vs. 56.78% vs. 53.66%, *p* = 0.006). Pregnant women receiving two doses of the COVID-19 vaccine demonstrated a better SBR against SARS-CoV-2 Alpha, Beta, and Gamma variants than women receiving just a single dose. Women who received the Tdap/flu vaccines demonstrated a better SBR when two COVID-19 vaccine doses were < 6 weeks apart. A better SBR was detected among women who had more recently received their second COVID-19 vaccine dose. Two doses of the COVID-19 vaccine provided recipients with a better SBR for Alpha/Beta/Gamma variants. Although Tdap/flu vaccines may affect the efficacy of the COVID-19 vaccine, different vaccination timings can improve the SBR.

## 1. Introduction

Since 2019, the Coronavirus disease 2019 (COVID-19) pandemic has spread life-threatening illness around the world [1]. COVID-19, caused by severe acute respiratory syndrome coronavirus 2 (SARS-CoV-2), has demonstrated high transmissibility and a global impact on health and the economy [1]. As of March 2022, more than 400 million people have been infected, and 6 million people have died from the disease [2]. The advent of the COVID-19 vaccine and the promotion of vaccination can greatly control the spread of the disease, reducing the incidence of severe illness and death related to COVID-19 [3], especially for those vulnerable cohorts, which include pregnant women, the elderly, and those with chronic disease history [4].

Pregnant women are especially impacted by the availability and use of COVID-19 vaccines, which prevent SARS-CoV-2 infection and subsequent maternal or neonatal morbidity. The literature supports the fact that pregnant women experience higher rates of COVID-19-associated severe illness or critical disease than the general population (23% vs. 19%) [5]. Another meta-analysis study summarizing 42 studies of 438,548 pregnant women showed higher rates of preeclampsia (OR 1.33, 95% CI 1.03–1.73), preterm birth (OR 1.82, 95% CI 1.38–2.39), stillbirth (OR 2.11, 95% CI 1.14–3.90), ICU admission (OR 4.78, 95% CI 2.03 to 11.25), and NICU admission (OR 3.69, 95% CI 1.39 to 9.82) among COVID-19-infected pregnant women compared to those without infection [6]. Further, pregnant women with symptomatic COVID-19 had a higher risk for Cesarean delivery and preterm birth compared to those with asymptomatic COVID-19 infections [6]. An increased risk of maternal and neonatal morbidity has also been reported in other publications [7,8,9].

There are no currently available COVID-19 vaccines for newborns. In our previous research, transplacental transmission of neutralizing antibodies was identified in the umbilical cord blood of newborn babies from vaccinated pregnant women, and vaccine protection with innate immunity to wild-type and delta-type SARS-CoV-2 was found in both mothers and babies [10]. The acquired antibody protection for neonates by the transplacental transmission of the maternal antibody can also be detected in pregnant women vaccinated with influenza (flu) vaccine as well as tetanus toxoid, reduced diphtheria toxoid, and acellular pertussis (Tdap) vaccines, without harmful impacts [11]. For these reasons, it is very important for pregnant women to receive the COVID-19 vaccination so that both mother and child can be protected. In addition, there has been plenty of research regarding both vaccine effectiveness and immunization in pregnant women [12,13,14,15,16,17].

Regarding protection from wild-type SARS-CoV-2 infection, neutralizing antibody protection achieved via the COVID-19 vaccine against SARS-CoV-2 variants including Alpha (B.1.1.7), Beta (B.1.351), Gamma (P.1), Delta (B.1.617.2), and Omicron (B.1.1.529) is reduced, compared to protection against wild-type infection, and breakthrough SARS-CoV-2 infections after vaccination have been reported [18]. The aim of our study was to investigate the effects of the COVID-19 vaccination on the binding ratio of anti-SARS-CoV-2 spike receptor binding domain IgG antibody (SRBD IgG) against Alpha-type, Beta-type, and Gamma-type SARS-CoV-2 among vaccinated pregnant women and their delivery neonates. We also studied the impact of the COVID-19 vaccination on the SRBD IgG binding ratio (SBR) among mothers and newborns when mothers had additionally received the flu and Tdap vaccines.

## 2. Materials and Methods

### 2.1. Study Design

This prospective study was performed at Kaohsiung Medical University Hospital. All participant collection and study protocols were approved by the local institutional review board (IRB) (IRB number: KMUHIRB-SV(II)-20210087), an ethics committee. All subjects had singleton pregnancies and were enrolled during hospital admission before delivery. Further, every enrolled mother was confirmed negative for SARS-CoV-2 infection by polymerase chain reaction test via nasopharyngeal swab before hospitalization, and none demonstrated symptoms or discomforts related to COVID-19 infection.

Every participant received their first dose of Moderna (mRNA-1273) vaccine after the first trimester of pregnancy, and 32 of them received a second dose of the COVID-19 vaccine during the 20th and 36th week of gestation. In addition to COVID-19 vaccination, 10 participants received 1 dose of Tdap vaccine (Adacel, Sanofi Pasteur, Toronto, ON, Canada) after the 28th week of gestation, 8 cases received flu vaccine (AdimFlu-S, QIS, Adimmune Corporation, Taichung, Taiwan; FlucelvaxQuad, CSL Behring GmbH, Marburg, Germany; VAXIGRIP TETRA, Sanofi Pasteur, Val-de-Reuil, Cedex, France) during pregnancy, and another 8 mothers received both Tdap and flu vaccine. Based on the various vaccination combinations, we classified the enrolled cases as: (1) group A: 2 doses of COVID-19 vaccine with Tdap and flu vaccination; (2) group B: 2 doses of COVID-19 vaccine with Tdap vaccination; (3) group C: 2 doses of COVID-19 vaccine with flu vaccination; (4) group D: 2 doses of COVID-19 vaccine; (5) group E: 1 dose of COVID-19 vaccine with Tdap vaccination; and, (6) group F: 1 dose of COVID-19 vaccine alone.

### 2.2. Patient Selection

All participant qualification was confirmed according to the inclusion and exclusion criteria: (1) age 20 years or older; (2) absence of prior COVID-19 vaccination and prior SARS-CoV-2 infection; (3) absence of preterm labor; (4) absence of any chronic medical disease history that required immunosuppressant treatment; (5) absence of any history of malignancy that required anti-cancer treatment.

### 2.3. Collection of Samples and Variables

Pregnant women meeting the above inclusion criteria were qualified for enrollment in our study while preparing for delivery and before hospitalization. During delivery, we collected the pregnant mother’s peripheral blood plasma and the delivery neonate’s umbilical cord blood plasma after umbilical clamping. The collected samples were then sent to the laboratory for further examination.

Other patient clinical variables, i.e., maternal age, maternal parity, maternal body weight mass index (BMI), neonatal body weight, neonatal gender, COVID-19 vaccination date, Tdap vaccination date, flu vaccination date, and gestational weeks of delivery were recorded for statistical analysis.

### 2.4. SRBD IgG Binding Ratio Test of SARS-CoV-2 Alpha, Beta, and Gamma Variants

SRBD IgG binding ratio for different SARS-CoV-2 variants following COVID-19 vaccination was detected by neutralizing antibody (Nab) inhibition test via enzyme-linked immunosorbent assay (ELISA). Spike protein receptor binding domain (SRBD) can be bound by neutralizing antibody (Nab) as well as angiotensin-converting enzyme 2 (ACE2). Because of this competitive inhibition binding, we detected the Nab titer from the mixture of collected samples by using an SRBD-containing solution and ACE2-coated wells and by comparing colorimetric changes. Nab-SRBD conjunction produced more colorimetric change compared to ACE2-SRBD conjunction.

We used a commercially available ELISA kit to detect Nab inhibition ratio for different SARS-CoV-2 variants including Alpha variant (anti-SARS-CoV-2, B.1.1.7, Neutralizing Antibody Titer Serologic Assay Kit, AcroBiosystem, Newark, DE, USA; Cat.: RAS-N028), Beta variant (anti-SARS-CoV-2, B.1.351, Neutralizing Antibody Titer Serologic Assay Kit, AcroBiosystem, Newark, DE, USA; Cat.: RAS-N031), and Gamma variant (anti-SARS-CoV-2, P.1, Neutralizing Antibody Titer Serologic Assay Kit, AcroBiosystem, Newark, DE, USA; Cat.: RAS-N034), and further experiments were performed based upon the manufacturer’s instructions.

The 96-well microplates were pre-coated with human ACE2 protein. Targeted samples, positive control, and negative control were added to the wells, followed by the addition of a solution containing horseradish peroxidase (HRP)-conjugated SRBD for the different SARS-CoV-2variants. Microplates were subsequently incubated in the dark for 1 h at room temperature. After incubation, supernatant was removed, and wells were washed 3 times with wash buffer. Additional substrate solution was added to wells, and microplates were incubated in the dark for 20 min at room temperature before stop solution was added to terminate the reaction. The result was a visible color change from blue to yellow. Colorimetric change in each well was evaluated by reading the O.D. value absorbance at 450 nm with a microplate spectrophotometer (Molecular Devices, USA). 

The obtained O.D. value results were used to calculate Nab inhibition percentage. More SRBD-Nab binding produced lower detected O.D. values. The calculation was based on the following formula:$$\mathrm{Inhibition}\;\%\;=\left(\;1-\frac{\mathrm{OD}450\;\mathrm{value}\;\mathrm{of}\;\mathrm{sample}}{\mathrm{average}\;\mathrm{OD}450\;\mathrm{value}\;\mathrm{of}\;\mathrm{negative}\;\mathrm{control}}\right)*100\%$$

The neutralizing antibody inhibition ratio with the commercially available ELISA kits that we used in this study can be used to evaluate the neutralizing antibody inhibition ratio, as above described, which mainly detected the SRBD IgG binding ratio. From the literature, the vaccine effectiveness can be assessed with the evaluation of the neutralizing antibody inhibition ratio in patients [19,20]. Using the ELISA kits that we used in this study to evaluate both vaccine effectiveness and immunization have been extensively utilized in much research [10,21,22,23,24]. In our study, we have attempted to obtain the binding ratio of SRBD IgG to spike protein from different variants in pregnant women and their neonates after vaccination as an assessment tool to investigate the effectiveness of the mRNA vaccine for these vulnerable cohorts.

### 2.5. Statistics

We analyzed the transplacental transmission of SRBD IgG binding ratio from mothers to infants using the neonatal–maternal ratio of Nab inhibition ratio determined by comparing results from neonatal cord blood divided by the results from maternal plasma. A sample t-test was performed to compare the SBR in maternal plasma, neonatal cord blood, and the transplacental transmission among different cohorts receiving different vaccine combinations, such as 1 or 2 doses of COVID-19 vaccine with or without Tdap or flu vaccination. Among subjects that received 2 COVID-19 vaccine doses, the same comparisons were also performed to analyze the SBR performance associated with the COVID-19 vaccine dose interval and the interval between final dose and delivery. The data were analyzed using GraphPad Prism (GraphPad Software, San Diego, CA, USA) and SPSS Statistics (version 22, IBM Corporation, Armonk, New York, USA), and a result of *p* < 0.05 was considered to be statistically significant.

## 3. Results

### 3.1. Participant Characteristics

A total of 39 eligible participants allowed us to obtain 39 samples of both maternal plasma and neonatal cord blood. Participant characteristics are listed in Table 1. Groups A, B, C, and D comprised 8 participants, and groups E and F comprised 2 and 5 participants, respectively. The mean age of total cohorts was 32.1 years of age (interquartile range, IQR 29–35), the mean maternal BMI was 26.8 (IQR 23.3–28.9), and the mean neonatal body weight was 3038.8 gm (IQR 2782.5–3170.0). The mean gestational age of delivery was 38.5 weeks (IQR 38.0–39.0), and the mean gestational weeks of the first and second doses of COVID-19 vaccines were 24.9 weeks (IQR 21.0–27.3) and 28.9 weeks (IQR 26.0–31.0), respectively.

Among participants receiving two full doses of the COVID-19 vaccine, the mean interval between doses was 5.3 weeks (IQR 4.0–6.0), and the median interval between the second dose and childbirth was 9.3 weeks (IQR 6.5–12.0). There were no significant differences for any of the above variables, however the time point for single vaccination dosing (30–31 weeks for participants in groups E and F) was later than the time point for the first of dual vaccination doses (22–25 weeks for all other participants). The *p* value among groups was 0.001).

### 3.2. SRBD IgG Binding Ratio for SARS-CoV-2 Alpha, Beta, and Gamma Variants

Appendix A shows the mean SBR for maternal plasma, neonatal cord blood, and the neonatal–maternal ratio for Alpha, Beta, and Gamma SARS-CoV-2 variants among all cohorts and for each group. Higher SBRs were found in maternal plasma and cord blood against Alpha variants compared to other variants (Alpha vs. Beta vs. Gamma variants: maternal plasma, 63.95% vs. 47.91% vs. 43.48%, *p* = 0.0001; cord blood, 72.14% vs. 56.78% vs. 53.66%, *p* = 0.006), but each of these SBRs was lower than the rate against wild-type SARS-CoV-2, which was over 90% according to previous research [10]. The trend toward a higher SBR for Alpha variants can also be found among participant groups that received two doses of the COVID-19 vaccine (groups A, B, C, and D: maternal plasma, *p* = 0.008–0.018; cord blood: *p* = 0.006–0.116). However, participants that received only a single COVID-19 vaccine dose demonstrated a different SBR. Among single-dose participant groups, group E demonstrated no differences in protection from any of the three variants (maternal plasma: *p* = 0658, cord blood: *p* = 0.795), and group F demonstrated a better SBR against the Beta variant compared to other types (Alpha vs. Beta vs. Gamma: cord blood, 15.42% vs. 44.97% vs. 11.43%, *p* = 0.040).

### 3.3. SRBD IgG Binding Ratio and Other Non-COVID Vaccine Combinations

SBR comparisons for all groups are provided in Appendix A. Among all participant groups, those that received two full doses of the COVID-19 vaccine (groups A, B, C, and D) demonstrated a higher SBR in maternal plasma and neonatal cord blood for every SARS-CoV-2 variant compared to participants that received only one dose of the COVID-19 vaccine (groups E and F) (Alpha type: maternal plasma as *p* = 0.006, cord blood as *p* = 0.002; Beta type: maternal plasma as *p* = 0.093, cord blood as *p* = 0.189; Gamma type: maternal plasma as *p* = 0.011, cord blood as *p* = 0.001). These differences are also illustrated in Figure 1, Figure 2, and Figure 3. What can also be taken from these figures is the fact that participants that received two full doses of the COVID-19 vaccine and that also received either the Tdap or flu vaccines demonstrated a decreased SBR against Alpha, Beta, and Gamma SARS-CoV-2 variants, as evidenced by maternal plasma and neonatal cord blood sample results. As shown in Appendix A, there were no significant SBR differences among those participants that received two full doses of the COVID-19 vaccine and both the Tdap and flu vaccines compared with those that received only two full doses of the COVID-19 vaccine. Additionally, there were no significant SBR differences between those participants that received two full doses of the COVID-19 vaccine and both the Tdap and flu vaccines and those participants that received two full doses of the COVID-19 vaccine and either the Tdap or flu vaccines. This decrease in the SBR was evident among participants receiving the Tdap vaccine (maternal: 67.56% vs. 85.57%, *p* = 0.051; neonatal: 79.73% vs. 89.75%, *p* = 0.088), as well as the flu vaccine (maternal: 68.93% vs. 85.57%, *p* = 0.044; neonatal: 80.33% vs. 89.75%, *p* = 0.022), but no such differences were detected for Beta or Gamma variants. There were no differences among those receiving two doses of COVID-19 vaccines with Tdap or flu vaccines.

### 3.4. SRBD IgG Binding Ratio in Different Intervals between Two Doses of COVID-19 Vaccines

We also compared the SBR in maternal plasma and neonatal cord blood for different variants using samples from participants that received both COVID-19 vaccine doses at different intervals, and the results are shown in Table 2. There were no significant differences in the SBR for the three variants among the 4-, 6-, or 8-week interval samples. Comparisons of the SBR for vaccine combinations for various intervals are provided in Appendix A. For participants with 4-week COVID-19 dose intervals, the SBR for most vaccine combinations demonstrated no differences, but groups receiving combination vaccinations that included two doses of the COVID-19 vaccine, the Tdap, and the flu vaccines demonstrated a better SBR against Beta and Gamma SARS-CoV-2 variants compared with those that received two doses of the COVID-19 vaccine and the flu vaccine alone (Beta: maternal plasma as 46.95% vs. 5.77%, *p* = 0.017; Gamma: cord blood, 65.18% vs. 27.89%, *p* = 0.068). The addition of the flu vaccine for participants that received two doses of the COVID-19 vaccine produced a lower SBR compared with those that received only two doses of the COVID-19 vaccine for Beta variants (maternal plasma, 5.77% vs. 78.18%, *p* = 0.055). Further, the receipt of two doses of the COVID-19 vaccine alone provided a better SBR compared with receiving two doses of the COVID-19 vaccine and the Tdap vaccine or compared with receiving two doses of the COVID-19 vaccine and the flu vaccine for the Beta and Gamma variants (Beta: *p* = 0.073–0.097; Gamma: *p* = 0.008–0.009).

Unlike results for the 4-week interval, there were no significant differences for Beta and Gamma variants among participants receiving two doses of the COVID-19 vaccine during the 4–6-week interval, but a better SBR was observed for those that received three vaccines compared with those that received the COVID-19 vaccine plus the Tdap vaccine (cord blood: 93.82% vs. 78.05%, *p* = 0.083) or the flu vaccine (cord blood: 93.82% vs. 75.86%, *p* = 0.019), and adding the flu vaccine to the COVID-19 vaccine also produced a lower SBR compared with COVID-19 vaccination alone (cord blood: 75.86% vs. 88.25%, *p* = 0.045). For those with dose intervals over 6 weeks, there were still no significant differences for Beta and Gamma variants, but those participants that received three vaccines had poor SBRs compared to those that received two doses of the COVID-19 vaccine alone (cord blood: 75.76% vs. 92.45%, *p* = 0.028) or with the Tdap (cord blood: 75.76% vs. 89.63%, *p* = 0.063) or flu vaccinations (maternal plasma: 60.93% vs. 78.73%, *p* = 0.088; cord blood: 77.58% vs. 92.45%, *p* = 0.016) for the SARS-CoV-2 Alpha variant.

### 3.5. SRBD IgG Binding Ratio for Different Intervals between Second COVID-19 Vaccine and Childbirth

Table 3 shows the SBR in maternal plasma and cord blood against different variants of SARS-CoV-2 when examining different intervals of time between the receipt of the second dose of the COVID-19 vaccine and childbirth. A better SBR against Alpha and Beta was detected for those with vaccine–childbirth intervals of less than 6 weeks compared with those experiencing vaccine–childbirth intervals of over 6 weeks (Alpha: maternal plasma, 85.17% vs. 69.34%, *p* = 0.090; Beta: maternal plasma, 73.86% vs. 48.89%, *p* = 0.049). A better SBR can be detected among those with vaccine–childbirth intervals of less than 8 weeks compared with those with vaccine–childbirth intervals over 8 weeks for Beta (maternal plasma: 67.86% vs. 44.48%, *p* = 0.015) and Gamma variants (maternal plasma: 62.15% vs. 43.99%, *p* = 0.071). Vaccine–childbirth intervals of less than 10 weeks were found to provide a better SBR than intervals over 10 weeks for Beta variants (maternal plasma: 62.38%, vs. 42.39%, *p* = 0.033; cord blood: 67.17% vs. 54.16%, *p* = 0.084). However, as listed in Appendix A, vaccine–childbirth intervals of over 12 weeks were associated with a better SBR compared with intervals less than 12 weeks for the Alpha variant (maternal plasma: 86.48%, vs. 68.35%, *p* = 0.001; cord blood: 90.79% vs. 80.55%, *p* = 0.032). In general, the SBR for the Alpha variant was poorest among the 8–10-week vaccine–childbirth interval, but protection increased after 10 weeks, and the best protection was associated with vaccine-childbirth intervals of less than 6 weeks and greater than 12 weeks (*p* = 0.023). No other obvious differences in the SBR were detected for Beta and Gamma variants.

SBRs based on different vaccine–childbirth intervals are displayed in Appendix A (interval ≤ 6 weeks), Appendix A (interval 6–8 weeks), Appendix A (interval 8–10 weeks), Appendix A (interval 10–12 weeks), Appendix A (interval over 12 weeks), Appendix A (interval over 6 weeks), and Appendix A (interval ≤ 12 weeks). Regarding intervals ≤ 6 weeks, adding the flu vaccine to those who have received two COVID-19 vaccines with or without the Tdap vaccine can increase the SBR for SARS-CoV-2 Beta variants (cord blood: 90.62% vs. 65.01%, *p* = 0.094). Regarding intervals of 6–8 weeks, the combination of three vaccines provided a better SBR against Beta variants compared with the COVID-19 vaccine alone (maternal plasma: 84.03% vs. 77.71%, *p* = 0.081). However, the SBR was poorer for those that received the COVID-19 vaccine and the Tdap vaccine compared with those that received the COVID-19 vaccine alone (maternal plasma: 57.75% vs. 77.71%, *p* = 0.026; cord blood: 58.28% vs. 83.31%, *p* = 0.039). Regarding the SBR for the Alpha variant, the COVID-19 vaccine plus the Tdap vaccination also provided less SBR than the COVID-19 vaccine plus the flu vaccination (Cord blood: 64.61% vs. 80.24%, *p* = 0.067). Regarding intervals of 8–10 weeks, participants receiving the COVID-19 vaccine and the flu vaccine demonstrated a better SBR against the Gamma variant compared with those that received the COVID-19 vaccine alone (maternal plasma: 48.78% vs. 8.73%, *p* = 0.014; cord blood: 72.23% vs. 27.14%, *p* = 0.048), but better results were found among those that received the COVID-19 vaccine and the Tdap vaccine (maternal plasma: 69.44% vs. 48.78%, *p* = 0.027). Regarding the 10–12 week interval, a better SBR against the Alpha variant was detected when the flu vaccination was added to those receiving the COVID-19 vaccine with or without the Tdap vaccination (cord blood: 82.40 % vs. 67.21%, *p* = 0.070), but the opposite result was found when the interval was greater than 12 weeks (maternal plasma: 78.82% vs. 90.31%, *p* = 0.030). Further, the use of the COVID-19 vaccine and the Tdap vaccine provided better SBR results compared with the COVID-19 vaccine and the flu vaccine for the Alpha variant among participants whose interval was greater than 12 weeks (maternal plasma: 90.0% vs. 78.82%, *p* = 0.076). For those whose interval was greater than 6 weeks, a three-vaccine combination provided a poorer SBR compared with the COVID-19 vaccination alone (maternal plasma: 63.18% vs. 82.23%, *p* = 0.071), and similar results were found among those who received the COVID-19 vaccine and the flu vaccination (cord blood: 80.33% vs. 88.94%, *p* = 0.096). Regarding the ≤ 12-week interval, those that received three vaccines demonstrated a better SBR compared with those that received the COVID-19 vaccine and the Tdap vaccine (cord blood: 81.87% vs. 70.83%, *p* = 0.096), but those that received the COVID-19 vaccine and the Tdap or flu vaccines demonstrated a better SBR compared with those that received the COVID-19 vaccination alone (COVID-19 vaccine plus Tdap vaccine: *p* = 0.002–0.003; COVID-19 vaccine plus Flu vaccine: *p* = 0.065–0.093).

## 4. Discussion

COVID-19 disease has threatened individual health and national economies worldwide. The development of vaccines and the promotion of vaccination have brought about highly impactful, positive change. The vaccine’s advent was especially important for pregnant women, and the transplacental transmission of Nab generated from vaccinated mothers to fetuses can also benefit newborns by providing protection from infections including SARS-CoV-2 [25,26,27,28,29]. From our previous research, we demonstrated the transplacental transmission of Nab inhibition against the wild-type and Delta SARS-CoV-2 variants with a neonatal-to-maternal ratio of 0.99 (0.99–1.00) and 0.90 (0.78–0.95), respectively [10]. In our current study, we also demonstrated the neonatal-to-maternal SRBD IgG ratio of approximately 1.18, 1.52, and 1.54 for Alpha, Beta, and Gamma SARS-CoV-2 variants, respectively. As with our previous study [10], the full two doses of the COVID-19 vaccination provided a significantly better SBR compared with a single dose against the Alpha (maternal plasma: 85.57% vs. 18.78%, *p* = 0.006; cord blood: 89.75% vs. 15.43%, *p* = 0.002), Beta (maternal plasma: 58.09% vs. 38.38%, *p* = 0.093), and Gamma variants (maternal plasma: 53.36% vs. 15.66%, *p* = 0.011; cord blood: 62.81% vs. 11.43%, *p* = 0.001). Interestingly, these SBRs were all less than wild-type results, which were over 90% in our previous research [10]. Moreover, better results were found for the SBR against the Alpha-type than the Beta or Gamma variants (Alpha vs. Beta vs. Gamma: maternal plasma, 63.95% vs. 47.91% vs. 43.48%, *p* = 0.0001; cord blood, 72.14% vs. 56.78% vs. 53.66%, *p* = 0.006). These results indicate that vaccine efficacy may be decreased with the immune escape of the virus as SARS-CoV-2 generates genetic mutations over time.

The Tdap vaccine is composed of tetanus toxoid, diphtheria toxoid, and acellular pertussis antigens [30]. The flu vaccine contains extracted hemagglutinin (HA) and neuraminidase (NA) antigenic proteins from the influenza virus that are further purified and concentrated for vaccine production [31]. Both the Tdap and flu vaccines are inactivated vaccines containing an infective microorganism subunit and are safe for pregnant women. From previous research, the administration of Tdap and flu vaccines for pregnant women can not only produce innate antibodies in mothers but also transfer the maternal antibody to the fetus by transplacental routes [11]. Additionally, maternal antibodies can also be detected in breast milk and can be transferred to babies during breastfeeding [32]. Therefore, pregnant women have been encouraged to receive the flu and Tdap vaccines, with Tdap vaccination advised during gestational age at 27–36 weeks to maximize maternal antibody response and passive transportation to the fetus [33,34]. Further, the flu vaccine can reduce systemic inflammation by reducing IL-6 production and downregulating toll-like receptor signaling pathways, Caspases protein, and FAS-associated death domain (FADD) that can adjust the response to SARS-CoV-2 infection and also decrease the COVID-19 burden [35,36]. The Tdap vaccine is a good inducer of cross-reactive CD8 T cells, and can also have a cross-reactive B cell reaction with the spike RBD domain, so the antibody produced by the Tdap vaccine can also inhibit viral engagement with ACE2 [37]. The Tdap vaccine may therefore enhance immunity against SARS-CoV-2 infection and further improve the COVID-19 disease outcome by relieving disease severity, especially for elderly and high-risk cohorts [38,39].

It is very important for the fetus and newborns to receive maternal antibodies as innate immunity to protect them from viral or bacterial infections because there are no available vaccines for these age cohorts. Because transplacental transmission of maternal antibodies to neonates has been identified among pregnant women receiving COVID-19 vaccination, it was interesting to determine the potential impacts of COVID-19 vaccination in pregnant mothers and additionally evaluate the transplacental transmission of maternal antibodies generated from the Tdap or flu vaccines. Among the participants receiving different vaccine combinations in this study, there were no obvious differences in SBR against Beta or Gamma variants, but the SBR for the Alpha variant in maternal plasma and cord blood were decreased when adding Tdap or flu vaccines for those who had received two full doses of the COVID-19 vaccine (add Tdap: maternal plasma, 67.56% vs. 85.57%, *p* = 0.051; cord blood, 79.73% vs. 89.75%, *p* = 0.088; add flu: maternal plasma, 68.93% vs. 85.57%, *p* = 0.044; cord blood, 80.33% vs. 89.75%). Although differences for Alpha variants were evident, the SBR was still over 50% for different vaccine combinations and appears to provide a safe and feasible route for providing protection against SARS-CoV-2 infection.

Delayed immunogenicity has been associated with the co-administration of the Tdap vaccine and the COVID-19 vaccine, but sufficient plasma Nab titer for SARS-CoV-2 can be found with simultaneously sufficient tetanus anti-toxoid [40]. Although no previous reports were found among COVID-19 vaccine trials [41], the co-administration of other vaccines with the COVID-19 vaccine was allowed [42]. The Tdap vaccine, as a glycoconjugate vaccine with carrier proteins, may cause an immunity interaction with other glycoconjugate vaccines including the CRM197-conjugated 13-valent pneumococcal vaccine (PCV13) and the TT-conjugated quadrivalent meningococcal vaccine (MCV4) due to antigen similarity [43,44,45]. For this reason, the delayed administration of the Tdap vaccine is recommended when the administration of multiple vaccinations is required as a means of avoiding immunogenicity interactions [46]. From our previous report, the concomitant vaccination of the flu vaccine and the COVID-19 vaccine is safe and preserves adequate immune reaction for both vaccines [47]. Although the co-administration of both vaccines may interfere with the immunity priming and may decrease the initial immunogenicity of the COVID-19 vaccine, there was no obvious impact on the subsequent immune reaction, and the co-administration of the flu vaccine with the second dose or the booster dose of the COVID-19 vaccine was therefore considered a suitable and safe procedure [48]. The research literature has also reported on the co-administration of the flu vaccine and the pneumococcal vaccine, and some serotypes were found with decreased immunogenicity, but no obvious impact on the immune reactions of the flu vaccine was reported [49].

The impact of different intervals between two COVID-19 vaccine doses on Nab protection has also been previously investigated. A delayed second dose may provide better vaccine coverage, which is especially relevant in regions with limited vaccine supplies, and overall mortality may be decreased by using a delayed second dose strategy [50,51]. Previously published literature also shows that a higher Nab titer can be found when the vaccine administration interval is extended to 6–14 weeks compared with 3–4 weeks, and better CD4 T cell with IL-2 expression has also been detected. A similar immune amplification response can also be seen in those with previous SARS-CoV-2 infection. For these reasons, dose interval extension is preferred and can generate better immunogenicity for COVID-19 vaccines [52]. Grunau et al. also reported that more viral Nab geometric mean value could be detected when employing long intervals (100–120 days) between two vaccine doses compared to short intervals (≤36 days) (302.3 vs. 41.8, *p* < 0.001) [53]. From our current study, we did not observe obvious differences across dosage intervals among all cohorts. No obvious differences were observed for the SBR of the Alpha variant at the 4-week interval, but a reduced SBR against Beta and Gamma variants was detected when COVID-19 vaccines were combined with the flu vaccine. However, no differences for Beta or Gamma variants were seen when the interval was over 4 weeks. Although combining with Tdap and flu vaccines provided an improved SBR for Alpha variants, when the COVID-19 vaccine interval increased from 4 to 6 weeks, this effect was attenuated. Therefore, employing a dosing interval within 6 weeks for pregnant women requiring Tdap and flu vaccines may be prudent.

Previous research literature noted that different Nab titers were detected at intervals subsequent to the receipt of the second, final COVID-19 vaccine dose. Ibarrondo et al. reported that the anti-RBD IgG value was enhanced rapidly at 3–4 days after the second dose and reached a plateau at the 7th–8th day, but the value may be decreased for every 10 days thereafter (*p* = 0.001) [54]. Wheeler et al. reported that those who only receive a single dose of the COVID-19 vaccine demonstrated significantly lower anti-S1 antibody levels compared to the level detected at 45 days or 75 days after receipt of the second dose of the vaccine (64 vs. 1053 vs. 821, *p* < 0.001) [55]. Although anti-S1 antibody levels were lower at day 75 than at day 45, no significant difference was found (*p* = 1.0) [55]. Interestingly, the anti-S2 antibody titer was also significantly increased after two doses of the COVID-19 vaccine, but the differences could only be seen after day 75 (*p* < 0.001) instead of after day 45 (*p* = 0.121), and no differences between the two detection timings was noted (*p* = 0.518) [55]. Naaber et al. reported a decrease in SRBD antibody level at 12 weeks and 6 months after receiving the second dose of the COVID-19 vaccine, and the value at the 6th month was only 7% of peak SRBD level, which was equal to those recovering from SARS-CoV-2 infection [56]. Additionally, this decrease was found for Beta and Gamma variants of SARS-CoV-2 [56]. Such antibody reduction may be predictable because the vaccine-induced plasma cell may not become a long-term memory plasma cell [57]. In our study, the SBR for the Beta and Gamma variants decreased with time, but the value was increased after 12 weeks for the Alpha type. Further, 6–8 weeks after the second dose, the SBR decreased when subjects also received the Tdap vaccine. During the 8–10-week interval, the SBR increased for the Gamma-type variant when participants had also received the flu vaccine. After 12 weeks, the SBR against Alpha type was better among those who had received the Tdap vaccine compared to those that received the flu vaccine, but that SBR decreased, regardless of whether or not the Tdap or flu vaccines were administered, at less than 12 weeks. As with other research [58,59], it is therefore recommended that the second dose vaccine–childbirth interval should not be too long and should take into account the Nab level induced by vaccination. 

Our study had some limitations. Following vaccine promotion, it was difficult to find pregnant women that could be vaccinated during pregnancy, so our study included only 39 participants and 78 samples, which may limit the value and impact of our results. No circulating variants of SARS-CoV-2 were subdivided in our current study. In addition, our participants only received the Moderna (mRNA-1273) vaccine, and only Alpha, Beta, and Gamma variants were investigated in our study, so the results may not represent all variants of SARS-CoV-2 under different vaccination schedules and combinations, and further study for other virus types may be needed in the future. From the literature, the surrogate virus neutralizing test (sVNT) used in our study can be a tool to assess and quantify the vaccine effectiveness via the evaluation of the ability of antibody production in our body [19,20]. The main limitation of sVNT was that it cannot recognize the quaternary epitopes in the S1 subunit and epitopes in S2. However, sVNT can be used to evaluate the neutralizing antibody inhibition ratio, meaning that this approach can be used to evaluate the vaccine effectiveness as well [10,21,22,23,24].

## 5. Conclusions

Pregnant women receiving two full doses of the COVID-19 vaccine demonstrated a better SRBD IgG binding ratio against Alpha, Beta, and Gamma SARS-CoV-2 variants. Between-dose intervals of less than 6 weeks may provide a better SBR for those requiring Tdap and flu vaccines. A shorter interval between the second dose of the COVID-19 vaccine and childbirth may provide a better SBR. Because the impact of vaccine co-administration was subtle, the COVID-19 vaccination is still considered suitable. Further studies and clinical trials with larger sample sizes are suggested for validation.

## Figures and Tables

**Figure 1 vaccines-10-01415-f001:**
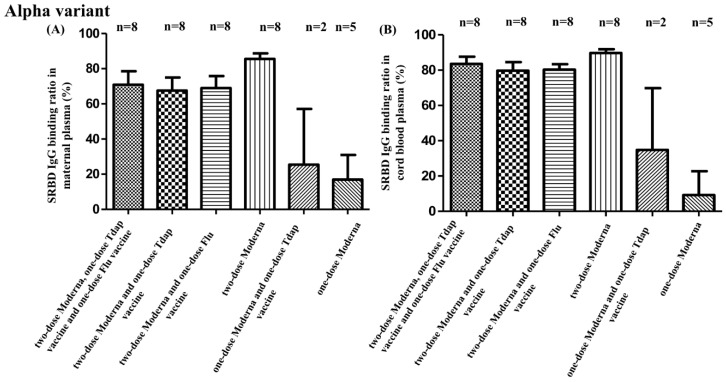
(**A**) SRBD IgG binding ratio against Alpha variants from maternal plasma in different vaccine combinations. (**B**) SRBD IgG binding ratio against Alpha variants from neonatal cord blood in different vaccine combinations. Tdap vaccine, tetanus toxoid, reduced diphtheria toxoid, and acellular pertussis vaccine; flu vaccine, influenza vaccine.

**Figure 2 vaccines-10-01415-f002:**
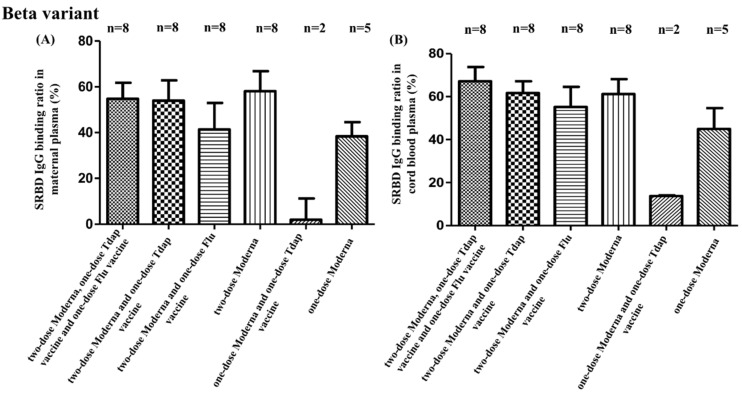
(**A**) SRBD IgG binding ratio against Beta variants from maternal plasma in different vaccine combinations. (**B**) SRBD IgG binding ratio against Beta variants from neonatal cord blood for different vaccine combinations. Tdap vaccine, tetanus toxoid, reduced diphtheria toxoid, and acellular pertussis vaccine; flu vaccine, influenza vaccine.

**Figure 3 vaccines-10-01415-f003:**
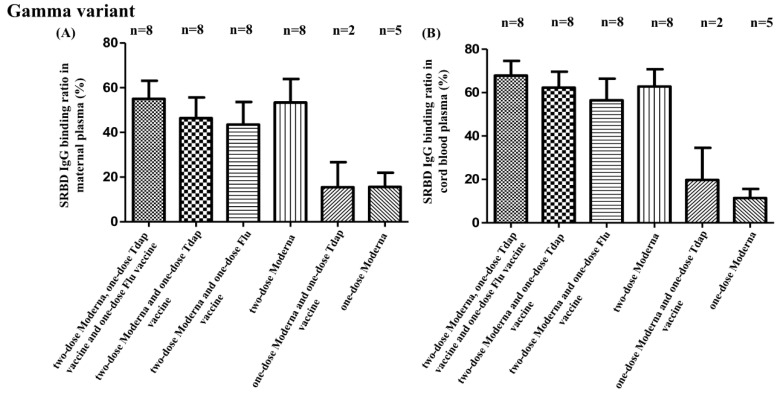
(**A**) SRBD IgG binding ratio against Gamma variants from maternal plasma in different vaccine combinations. (**B**) SRBD IgG binding ratio against Gamma variants from neonatal cord blood in different vaccine combinations. Tdap vaccine, tetanus toxoid, reduced diphtheria toxoid, and acellular pertussis vaccine; flu vaccine, influenza vaccine.

**Table 1 vaccines-10-01415-t001:** Patient Characteristics.

Median (IQR)	Total (N = 39)	Group A (N = 8)	Group B (N = 8)	Group C (N = 8)	Group D (N = 8)	Group E (N = 2)	Group F (N = 5)	*p* Value
Age (yrs) (IQR)	32.1 (29.0–35.0)	29.0 (28.25–31.0)	32.4 (29.0–34.75)	30.5 (26.75–33.75)	35.0 (28.75–41.5)	38.0 (35.0–NA)	32.0 (26–36.5)	0.133
Parity (%)								0.174
Primiparity	19 (48.71)	5 (62.5)	6 (75)	2 (25)	4 (50)	1 (50)	1 (20)	
Multiparity	20 (51.29)	3 (37.5)	2 (25)	6 (75)	4 (50)	1 (50)	4 (80)	
Baby body weight (g)	3038.8 (2782.5–3170.0)	3032.9 (2985–3150)	3057.5 (2810–3393.75)	2936.9 (2723.75–3111.25)	3153.8 (2700–3575)	327.0 (3135–NA)	2902.0 (2690–3082.5)	0.664
Maternal BMI	26.8 (23.3–28.9)	29.5 (25.2–34.6)	26.1 (22.8–30.0)	25.2 (23.1–27.6)	26.3 (23.7–27.5)	31.6 (28.0–NA)	25.1 (22.5–27.8)	0.125
Baby gender (%)								0.480
Female	21 (53.8)	6 (75)	5 (62.5)	4 (50)	2 (25)	1 (50)	3 (60)	
Male	18 (46.2)	2 (25)	3 (37.5)	4 (50)	6 (75)	1 (50)	2 (40)	
Delivery GA (wks)	38.5 (38.0–39.0)	38.7 (38.0–40.0)	39.0 (38.3–39.8)	38.5 (37.3–39.0)	38.3 (38.0 –39.8)	38.0 (38.0–38.0)	38.2 (36.5–39.5)	0.756
1st COVID-19 vaccine GA (wks)	24.9 (21.0–27.3)	23.4 (22.0–25.0)	23.5 (21.3–26.0)	22.8 (21.0–24.8)	25.0 (21.8 –27.8)	31.5 (30.0–NA)	30.4 (25.5–35.0)	**0.001**
2nd COVID-19 vaccine GA (wks)	28.9 (26.0–31.0)	28.7 (26.0–31.0)	28.3 (26.3–30.8)	27.8 (25.3–29.8)	31.3 (28.0 –33.0)	-	-	0.182
Interval between 2 doses of COVID-19 vaccines (wks)	5.3 (4.0–6.0)	5.3 (4.0–7.0)	4.8 (4.0–5.0)	5.0 (4.25–5.0)	6.4 (5.0–8.0)	-	-	0.108
Interval between 2nd COVID-19 vaccine to baby delivery (wks)	9.3 (6.5–12.0)	10.0 (8.0–12.0)	10.8 (8.25–13.0)	10.8 (8.5–12.75)	7.4 (5.0–10.0)	6.5 (5.0–NA)	7.2 (4.0–11.0)	0.166

IQR, interquartile range; yrs, years; BMI, body mass index; GA, gestational age; wks, weeks; words in bold: *p* value with significance.

**Table 2 vaccines-10-01415-t002:** SRBD IgG binding ratio for SARS-CoV-2 Alpha, Beta, and Gamma variants with different intervals between two COVID-19 vaccine doses.

		Alpha Type				Beta Type				Gamma Type			
		Arm 1	Arm 2	*p* Value		Arm 1	Arm 2	*p* Value		Arm 1	Arm 2	*p* Value	
Arm 1: ≤ 4 weeks Arm 2: > 4 weeks	N MP (%) CB (%) Ratio	9 69.58 82.88 1.25	21 73.01 82.47 1.20	0.659 0.925 0.665		9 45.97 58.86 2.29	21 56.09 62.05 1.25	0.339 0.704 0.189		9 47.96 61.88 1.75	21 51.80 63.11 1.55	0.724 0.895 0.619	
Arm 1: ≤ 6 weeks Arm 2: > 6 weeks	N MP (%) CB (%) Ratio	24 71.39 81.99 1.22	6 74.33 85.01 1.21	0.741 0.543 0.931		24 49.80 59.43 1.69	6 66.06 67.78 1.07	0.176 0.383 0.303		24 47.59 60.47 1.65	6 62.88 71.81 1.43	0.213 0.282 0.635	
Arm 1: ≤ 8 weeks Arm 2: > 8 weeks	N MP (%) CB (%) Ratio	29 71.10 82.09 1.22	1 97.34 97.19 0.99	0.179 0.167 0.481		29 51.78 61.10 1.59	1 89.95 60.86 0.68	0.153 0.991 0.498		29 49.63 61.81 1.63	1 80.35 89.60 1.12	0.263 0.236 0.631	
		**Arm 1**	**Arm 2**	**Arm 3**	***p* Value**	**Arm 1**	**Arm 2**	**Arm 3**	***p* Value**	**Arm 1**	**Arm 2**	**Arm 3**	***p* Value**
Arm 1: ≤ 4 weeks Arm 2: 4–6 weeks Arm 3: > 6 weeks	N MP (%) CB (%) Ratio	9 69.58 82.88 1.25	15 72.48 81.46 1.19	6 74.33 85.01 1.21	0.892 0.795 0.911	9 45.97 58.86 2.29	15 52.10 59.76 1.33	6 66.06 67.78 1.07	0.348 0.685 0.122	9 47.96 61.88 1.75	15 47.37 59.63 1.59	6 62.88 71.81 1.43	0.467 0.552 0.839

MP, maternal plasma; CB, cord blood.

**Table 3 vaccines-10-01415-t003:** SRBD IgG binding ratio for SARS-CoV-2 Alpha, Beta, and Gamma variants with comparisons of different intervals from 2nd COVID-19 vaccine dose to childbirth.

Two Doses Vaccine		Alpha Type			Beta Type			Gamma Type		
Interval between Last Dose to Delivery		Arm 1	Arm 2	*p* Value	Arm 1	Arm 2	*p* Value	Arm 1	Arm 2	*p* Value
Arm 1: ≤ 6 wks Arm 2: > 6 wks	N MP (%) CB (%) Ratio	5 85.17 87.40 1.03	25 69.34 81.64 1.25	0.090 0.276 0.140	5 73.86 70.13 0.98	25 48.89 59.29 1.68	**0.049** 0.290 0.281	5 66.87 73.57 1.12	25 47.41 60.57 1.71	0.138 0.250 0.247
Arm 1: ≤ 8 wks Arm 2: > 8 wks	N MP (%) CB (%) Ratio	11 77.03 83.23 1.11	19 69.06 82.23 1.28	0.276 0.810 0.136	11 67.86 69.08 1.05	19 44.48 56.47 1.86	**0.015** 0.107 0.039	11 62.15 70.62 1.21	19 43.99 58.17 1.84	0.071 0.152 0.041
Arm 1: ≤ 10 wks Arm 2: > 10 wks	N MP (%) CB (%) Ratio	16 71.72 81.22 1.20	14 72.27 84.17 1.23	0.939 0.457 0.816	16 62.38 67.17 1.19	14 42.39 54.16 1.99	**0.033** 0.084 0.122	16 56.97 68.64 1.37	14 43.42 55.99 1.88	0.168 0.131 0.196
Arm 1: ≤ 12 wks Arm 2: > 12 wks	N MP (%) CB (%) Ratio	24 68.35 80.55 1.26	6 86.48 90.79 1.05	**0.001** 0.032 0.144	24 55.73 63.37 1.37	6 42.35 52.02 2.36	0.268 0.393 0.352	24 55.00 65.28 1.34	6 33.23 52.59 2.67	0.072 0.228 0.125

MP, maternal plasma; CB, cord blood; Words in bold: *p* value with significance.

## Data Availability

Not applicable.

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
