# Peer review of "Antibodies against SARS-CoV-2 Alpha, Beta, and Gamma Variants in Pregnant Women and Their Neonates under Antenatal Vaccination with Moderna (mRNA-1273) Vaccine"

_vaccines, 2022, doi:10.3390/vaccines10091415_

Round 1
Reviewer 1 Report (Previous Reviewer 2)
The manuscript entitled “Antibodies against SARS-CoV-2 alpha, beta, and gamma 2 variants in pregnant women and neonates vaccinated with Moderna (mRNA‐1273) vaccine.” by Chen et. al. is an interesting and well-written manuscript. The new title of the manuscript is more representative title of the study as compared to the earlier one. The only concern of the study is that this study is based on a very small sample size (78 samples from 39 pregnant women). However, authors tried to address most of the comments raised in this study and the suggested modifications have been taken into account in improving the quality of the article. The revised version of the manuscript is clear, concise, and well-written.
Author Response
Please see the attachment.

Reviewer 2 Report (Previous Reviewer 1)
The new title is confusing: the neonates were not vaccinated.
The added lines 153-154 do not clearly explain what SRBD is in plain English. Also this measurements do not give any information about the protective effect of the vaccine. The sVNT data referenced in the response to reviewers is not included in this manuscript. The only comments that can be made about these results are about antibody levels, nothing more. There correlates of protection in vaccination have not been identified as far as this reviewer knows so there is no direct correlation between the levels of anti-RBD IgG and vaccine efficiency. As recommend before the text needs to be edited to remove any link between antibody binding ration and protection due to vaccination.
From the response to reviewers: " The data about Delta and Omicron (BA.1 BA.2 BA.5) is currently analyzed and prepared writing an article. " Then add the Delta and Omicron data to this manuscript to strengthen it.
There are still missing references in this manuscript, to list a few:
Halasa et al. MMWR 2022, Carlsen et. al JAMA Network 2022, Prabhu et al. Obst. Gynecol. 2021, Beharier et al JCI 2021, Mithal et al. Am. J. Obst. Gynecol. 2021, Yang et al. Obst. Gynecol. 2021, Atyeo et al. Nat. Comm. 2022.
Round 2
Reviewer 2 Report (Previous Reviewer 1)
The current title "Antibodies against SARS-CoV-2 alpha, beta, and gamma variants pregnant women and their neonates under antenatal vaccination with Moderna (mRNA‐1273) vaccine " does not make sense. Maybe "Antibodies against ... variants in pregnant women and their neonates..."
Please add a sentence explaining that vaccine effectiveness in your manuscript is defined as the production of spike-binding antibodies.
Author Response
Please see the attachment.

This manuscript is a resubmission of an earlier submission. The following is a list of the peer review reports and author responses from that submission.
Round 1
Reviewer 1 Report
The authors have addressed some of the concerns raised by the reviewers.
line 152: change Inhibition % to SBR
Please explain the relevance of SBR and what it means in plain English at some point in the manuscript.
Reviewer 2 Report
The manuscript entitled “Anti-SARS-CoV-2 spike receptor binding domain antibody binding ratio for SARS-CoV-2 alpha, beta, and gamma variants in vaccinated pregnant women and neonates” by Chen et. al. is an interesting manuscript. However, there were some major concerns related to this study, such as a very small sample size (39 pregnant women at Kaohsiung Medical University Hospital), and also the authors were not able to clearly address some other important variants of SARS-CoV2 in this study such as Delta and Omicron variants. According to the authors, this paper is in leaving an academic report that future research on delta and omicron may be able to leverage moving forward. But, in my opinion, this information will be more impactful and valuable for the current vaccination strategy, where these new variants are threatening the worldwide population.
Reviewer 3 Report
Thank you for the revised manuscript. The original title of the article was easier to read. My point was that the ELISA method measures the binding antibodies, and the binding antibodies may or may not be the neutralising antibody, although it is likely that the antibodies that bind the spike are neutralising antibodies. Only by direct neutralising the virus in a containment laboratory, you can measure the neutralising antibodies. The manufacturer does claim the kits measure the neutralising antibodies, which is not strictly true.
Suggested title:
Antibodies against SARS-CoV-2 alpha, beta, and gamma 2 variants in pregnant women and neonates vaccinated with Moderna (mRNA‐1273) vaccine
I suggest making similar changes throughout.
I don’t understand why maternal plasma and cord blood were used (in abstract and throughout), except in line 116 cord blood plasma. Line 138, the manufacturer’s instruction is to use serum samples.
Table 1. The p values are only meaningful when the different treatments are applied to the randomly selected groups. You cannot select the group first and then calculate the p-values. If you want to show that different groups are comparable, p-values do not serve the purpose. Please seek statistical advice for your analyses.